# BaFTA: Backprop-free Test-Time Adaptation for Zero-Shot Visual Language Models

## Abstract

Large-scale pretrained vision-language models like CLIP have demonstrated remarkable zero-shot image classification capabilities across diverse domains. To enhance CLIP's performance while preserving the zero-shot paradigm, various test-time prompt tuning methods have been introduced to refine class embeddings through unsupervised learning objectives during inference. However, these methods often encounter challenges in selecting appropriate learning rates to prevent model instability with absence of validation data during test-time training. In this study, we propose a novel backpropagation-free method for test-time adaptation in vision-language models. Instead of fine-tuning text prompts to refine class embeddings, our approach directly estimates class centroids using online clustering within a projected embedding space that aligns text and visual embeddings. We dynamically aggregate predictions from both estimated and original class embeddings, as well as from distinct augmented views, by assessing the reliability of each prediction using Rényi entropy. Through extensive experimentation, we demonstrate that our approach consistently outperforms state-of-the-art test-time adaptation methods by a significant margin.

## 1 Introduction

The emergence of large-scale pre-trained "foundation" vision-language models (VLM), exemplified by pioneering works such as CLIP (Radford et al. (2021)) and ALIGN (Jia et al. (2021)) has ushered in a new era of computational capabilities. These models have demonstrated promising capacity for open-world generalization, where their ability to excel in tasks extends beyond the original training data. They achieve this feat by harnessing an unified visual-text embedding space, enabling them to perform zero-shot image classification on novel concepts, simply by translating the category names into this shared representation as classification weights.

In the pursuit to further enhance VLM performance, various adaptation and fine-tuning techniques have emerged to bridge the domain gap when applied to downstream tasks. For instance, Zhou et al. (2022b) and Zhou et al. (2022a) fine-tune text prompts for VLMs, tailoring them to specific downstream tasks with few-shot adaptation. Moreover, in the realm of zero-shot classification, numerous approaches have been proposed to boost VLM performance without necessitating labeled data. For example, Hu et al. (2023) and Tanwisuth et al. (2023) improve CLIP through source-free unsupervised adaptation using unlabeled test examples, while Udandarao et al. (2022) and Ge et al. (2023) enhance CLIP with training-free methods by leveraging external resources. Furthermore, test-time prompt tuning algorithms, exemplified by Manli et al. (2022) and Park & D'Amico (2023), refine learnable text prompts during inference through the optimization of an unsupervised objective using augmentations, leading to improved model accuracy.

As shown in Table 1, the test-time prompt-tuning method, TPT (Manli et al. (2022)), excels in adaptation without the need for labeled data or external resources. This characteristic underscores its practicality and versatility as an effective means to enhance the performance of CLIP, especially within the context of zero-shot classification. However, as pointed out by Niu et al. (2023), test-time adaptation methods such as Wang et al. (2020) Liang et al. (2023) as well as methods like TPT, often encounter the intricate challenge of determining an optimal learning rate in the absence of validation data. Striking the right balance is crucial—achieving maximum improvement while simultaneously safeguarding against the model's instability during test-time adaptation.

| Methods | Task Setting | Labeled Data | Back-Prop Training | External Resource |
|---------|-------------|--------------|--------------------|--------------------|
| CoOp (Zhou et al. (2022b)) | Few-Shot Fine-Tuning | Yes | Yes | None |
| CoCoOp (Zhou et al. (2022a)) | Few-Shot Fine-Tuning | Yes | Yes | None |
| ReCLIP (Hu et al. (2023)) | Source-Free Adaptation | No | Yes | None |
| POUF (Tanwisuth et al. (2023)) | Source-Free Adaptation | No | Yes | None |
| TPT (Manli et al. (2022)) | Test-Time Adaptation | No | Yes | None |
| DiffTPT (Feng et al. (2023)) | Test-Time Adaptation | No | Yes | Stable Diffusion |
| Sus-X (Udandarao et al. (2022)) | Test-Time Adaptation | No | No | LAION-5B |
| Hierarchy-CLIP (Ge et al. (2023)) | Test-Time Adaptation | No | No | ImageNet Hierarchy |
| BaFTA (Ours) | Test-Time Adaptation | No | No | None |

Table 1: Taxonomy of CLIP adaptation methods for downstream classification. In this work, we adopt TPT as the main baseline for comparison as it is the state-of-the-art test-time adaptation algorithm without requirement of external resources.

To avoid the trouble in determining the optimal learning rate and to fully harness the potential of each test example while avoiding concerns about model instability, we propose the Backpropagation-Free Test-time Adaptation algorithm BaFTA. Instead of refining the class embeddings with backpropagation training in the prompt token space, BaFTA directly refines the class embeddings within the unified visual-text embedding space of CLIP, by leveraging the neighboring information among test examples visual embeddings with an online clustering algorithm. Our approach is motivated by the observation that the visual embeddings from CLIP are often sufficiently discriminative for effective classification. However, the sub-optimal zero-shot performance is often limited by the imprecise text embeddings associated with inaccurate class names. Therefore, we opt to harness neighboring information within test example visual embeddings to enhance CLIP's test-time performance.

To further enhance the performance of online clustering predictions, we have proposed two pivotal designs. Firstly, building upon the recommendation from Hu et al. (2023), we execute the online clustering algorithm on a projected embedding space. This projection helps alleviate the disparity between CLIP's visual and text embeddings, contributing to improved clustering outcomes. Secondly, recognizing that clustering-based predictions can sometimes be swayed by the biased distribution of test examples, we combine these clustering-based predictions with standard predictions derived from randomly augmented views of the test examples. We employ Rényi Entropy to gauge the reliability of these predictions, ultimately arriving at an aggregated prediction that benefits from the strengths of both approaches while ensuring accuracy and robustness.

The significance of our work can be summarized in four key contributions:

- We introduce BaFTA, a novel Backpropagation-Free Test-time Adaptation algorithm designed to enhance the zero-shot classification capability of vision-language models at inference time, without requiring any labeled examples or back propagation training.

- We propose an effective online clustering method to directly refine the class embeddings of vision-language models within a projected space that aligns the visual and text embeddings.

- We present a simple technique to dynamically aggregate the predictions from the clustering-estimated and original class embeddings, as well as from various augmented views, by evaluating the reliability of each prediction using Rényi entropy.

- Through comprehensive experiments, we validate BaFTA and its components, affirming its effectiveness in significantly improving the zero-shot classification accuracy of pre-trained vision-language models during inference.

## 2 BACKGROUND

In this section, we revisit the large-scale pre-trained vision language model CLIP (Radford et al. (2021)) and test-time prompt tuning algorithm TPT (Manli et al. (2022)) for the necessary background before we introduce our method in Section 3.

**Zero-Shot Image Classification with VLM.** A pre-trained vision-language model such as CLIP consists of two parallel components $M = \{M_v, M_t\}$ where $M_v$ is the visual encoder and $M_t$ is the

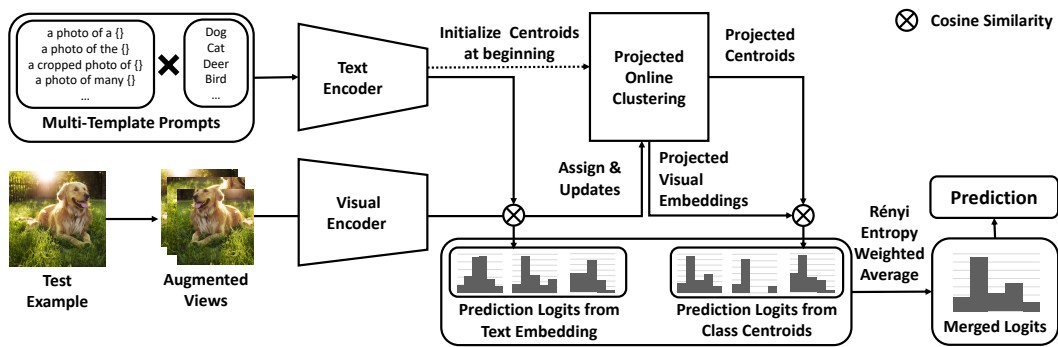

Figure 1: Overview of our Backpropagation-Free Test-Time Adaptation algorithm BaFTA. Instead of prompt-tuning, we employ online clustering to directly estimate class embeddings in a projection space that aligns visual and text embeddings. The class centroids are initialized with text embeddings of class names, and updated incrementally with online test examples assigned to the class. For each test example, we generate two sets of predictions. The first set measures cosine similarity between visual embeddings of augmented views and class name text embeddings. The second set measures cosine similarity between visual embeddings and online-clustering centroids. Predictions are aggregated with reliability estimated by Rényi Entropy for final results.

text encoder. Given test images $D^{test} = \{x_i\}_{i=1}^I$ and target class names $C = \{c_j\}_{j=1}^J$ , the pre-trained vision-language $M$ performs zero-shot classification by generating the adaptive classification weights from text embeddings of the target class names $t_j = M_t(\theta_0(c_j))$ for $j \in \{1, 2, ..., J\}$, where $\theta_0$ is the text prompt template such as "a photo of {class name}" that warped the class names $c_j$ into full sentences $\theta_0(c_j)$. To further improve the quality of the text embeddings, CLIP provides lists of templates $\{\theta_z\}_{z=1}^Z$ to align the text embeddings with the distribution of real caption sentences used in pre-training, and generates the text embeddings for each class by taking the average of these templates,

$$t_j = \frac{1}{Z} \sum_{z=1}^Z M_t(\theta_z(c_j)).$$

Then, the prediction $y_i$ can be obtained by selecting the class $j$ whose text embedding $t_j$ has the highest cosine similarity with its visual embedding $M_v(x_i)$, i.e., $y_i = \arg\max_j \langle \frac{M_v(x_i)}{\|M_v(x_i)\|}, \frac{t_j}{\|t_j\|} \rangle$

**Test-Time Prompt Tuning for VLM.** To further enhance the zero-shot generalization ability of vision language model $M$, TPT proposes to learn an adaptive text template $\theta$ at inference time. For each test example $x_i$, TPT first prepares a mini-batch of random augmented views $\{x_i^1, x_i^2, ..., x_i^B\}$ and performs a single step gradient descent to optimize the entropy minimization loss over the high-confidence predictions among the augmented views,

$$\theta_i = \theta_0 - \delta\nabla_\theta \left( \sum_{b=1}^B \mathbb{1}[H(M(x_i^b) < \tau]H(M(x_i^b)) \right)|_{\theta=\theta_0}$$

where $H(\cdot)$ is the entropy function, $\tau$ is the entropy threshold for high-confidence augmented view selection, and $\delta$ is the learning rate. $M(x_i^b) = softmax\left( [M(x_i^b; c_j)]_{j=1}^J \right)$ is the esti-mated probability distribution of augmented view $x_i^b$ over target classes $c_1, ..., c_j$, with $M(x_i^b; c_j) = \left\langle \frac{M_v(x_i^b)}{\|M_v(x_i^b)\|}, \frac{M_t(\theta(c_j))}{\|M_t(\theta(c_j))\|} \right\rangle$ as the cosine-similarity between visual embedding $M_v(x_i^b)$ and text embed-ding $M_t(\theta(c_j))$. Then, with adapted text prompt $\theta_i$, TPT produces the prediction for test example $x_i$ with the averaged prediction from high-confidence (estimated by entropy $H(\cdot)$) augmented views:

$$y_i = \arg\max_j \sum_{b=1}^B \mathbb{1}[H(M(x_i^b) < \tau]M(x_i^b; c_j) \qquad (1)$$

## 3 METHOD

As investigated by Niu et al. (2023), test-time adaptation algorithms frequently encounter the challenge regarding the appropriate choice of the learning rate in absence of validation data during unsupervised training. On one hand, opting for a small learning rate will restrict the enhancement of the model. On the other hand, employing a large learning rate can be risky in triggering the potential model collapse. TPT adopts a relatively large learning rate to expedite improvement, but chooses to restart from the original model for each test example to prevent the potential model collapse.

In this work, we present an novel backpropagation-free solution which directly refines the class embeddings in the aligned visual-text embedding space instead of in the prompt token space. Our BaFTA algorithm performs **Ba**ckpropagation-**F**ree **T**est-time **A**daptation for Vision-Language Models, and brings three major advantages over the test-time prompt tuning methods like TPT:

- BaFTA avoids the use of back-propagation to update model weights. As a result, it significantly reduces the risk of causing model collapse during unsupervised adaptation.
- In contrast to the test-time adaptation algorithms like TPT that require frequent restart to prevent model collapse, BaFTA possesses the capability to scale as more test examples become available, and to leverage from the relationships between neighboring examples.
- BaFTA can leverage the multi-template prompts provided by CLIP to enhance text embedding quality. In contrast, prompt-tuning methods are constrained to using single-template prompts due to computational costs.

In the following sections, we first present the motivation and primary concepts behind the estimation of class embeddings using online clustering during inference, as outlined in Section 3.1. Subsequently, we delve into the discussion of two pivotal findings that enhance the performance of online clustering, as elaborated in Section 3.2 and Section 3.3 respectively. Finally, we present a comprehensive overview of the BaFTA with the complete algorithm in Section 3.4.

### 3.1 ESTIMATE CLASS EMBEDDING WITH ONLINE CLUSTERING

As shown in Table 2, CLIP generates discriminative visual embeddings on various downstream tasks, but the zero-shot classification performance is often limited by the imprecise text embeddings generated from uninformative class names. For example, FGVC Aircraft ( Maji et al. (2013)) uses codenames such as `707-320` and `A300B4` as class names, which are hardly informative for CLIP to generate proper text embeddings to capture the visual difference between classes.

Conversely, the results of linear evaluation suggest that the visual embeddings from CLIP exhibit a high degree of distinctiveness among target classes, enabling the linear classifier to attain remarkable classification accuracy. This finding opens up an opportunity to leverage the neighboring information within these visual embeddings to further enhance classification performance.

Given a set of visual embeddings $\{v_i | v_i = M_v(x_i)\}_{i=1}^I$ come in order, we can obtain a set of cluster centroids $w_j$ as class embeddings using the online clustering algorithm Barbakh & Fyfe (2008):

$$w_j = \frac{t_j}{\|t_j\|} \qquad \text{initialize centroids with text embedding } t_j$$

$$w_{y_i} = \frac{k_{y_i} w_{y_i} + v_i}{\|k_{y_i} w_{y_i} + v_i\|} \qquad \text{update upon example } v_i \text{ with prediction } y_i \qquad (2)$$

$$k_{y_i} = k_{y_i} + 1 \qquad \text{update counter } k_{y_i} \text{ for class } y_i$$

where $k_{y_i}$ records the number of examples contributed to the calculation of $w_{y_i}$ before $v_i$, which adjust the magnitude of $w_{y_i}$ to accommodate the new cluster member $v_i$.

### 3.2 VISUAL TEXT ALIGNMENT

While VLMs aim to establish a unified embedding space for both visual and text modalities, recent research studies conducted by Liang et al. (2022), Tanwisuth et al. (2023) and Hu et al. (2023) have suggested that contrastive pre-trained models might still exhibit a notable disparity between

| | Cars | Caltech101 | DTD | EuroSAT | FGVC | Food101 | Flower102 | Pets | UCF101 | SUN397 | ImageNet |
|---|---|---|---|---|---|---|---|---|---|---|---|
| CLIP (RN50) *Zero-Shot* | 55.8 | 82.1 | 41.7 | 41.1 | 19.3 | 81.1 | 65.9 | 85.4 | 63.6 | 59.6 | 59.6 |
| CLIP (RN50) *Linear-Eval* | **78.3** | **89.6** | **76.4** | **95.2** | **49.1** | **86.4** | **96.1** | **88.2** | **81.6** | **73.3** | **73.3** |
| CLIP (ViT-B/16) *Zero-Shot* | 65.6 | 89.3 | 46.0 | 54.1 | 27.1 | 89.2 | 70.4 | 88.9 | 69.8 | 65.2 | 68.6 |
| CLIP (ViT-B/16) *Linear-Eval* | **86.7** | **94.7** | **79.2** | **97.1** | **59.5** | **92.8** | **98.1** | **93.1** | **88.4** | **78.4** | **80.2** |

Table 2: Zero-Shot v.s. Linear Evaluation top-1 accuracy reported by CLIP (Radford et al. (2021)). Linear Evaluation protocol assesses the quality of visual embeddings by training a fully-supervised linear classifier over the frozen visual embeddings. This Linear Evaluation result implies: **1)** the zero-shot performance of CLIP are largely limited by the quality of zero-shot classifier, i.e, the text embeddings of class names; **2)** The native visual embeddings of CLIP get classified well with a linear classifier, which suggests the distinctiveness of visual embeddings across target classes, and leads to an opportunity to leverage the neighboring relationships to enhance test-time performance.

their visual and text embeddings. Hu et al. (2023) introduces a simple yet effective projection-based alignment method. This method effectively removes the classification-agnostic information that is inherent in both visual and text modalities. As a result, it efficiently aligns the visual and text embeddings, leading to the advantages of enhanced embedding distribution and clustering characteristics.

Assuming a classification task with $J$ classes, where the text embeddings are denoted as $T = [t_1, ..., t_J]$ with $t_j = M_t(c_j)$. Using the singular value decomposition

$$U, S, V = svd(T)$$

we obtain $U = [e_1, e_2, ..., e_J]$ as the orthonormal basis of the span of $T$, that defines a matrix $P = UU^\top$. This matrix projects embeddings onto the span of $T$ and removes the class-agnostic information irrelevant to classification. Additionally, the principle axis $e_1$ within the outer space basis $U$ represents where $\{t_1, ..., t_J\}$ overlap the most. By removing $e_1$, the text embeddings are separated from each other, which in turn distances the visual embeddings of different classes. Together with feature re-normalization, Hu et al. (2023) defines the projection function $P^*$ with

$$P^*(x) := \frac{U'U'^\top x}{\|U'U'^\top x\|} \qquad\qquad U' = [e_2, e_2, ..., e_J] \qquad\qquad (3)$$

### 3.3 PREDICTION AGGREGATION WITH RÉNYI ENTROPY

The online clustering algorithm presented in Section 3.1 yields accurate estimations of the embedding centroids for classes that have a sufficient quantity of seen test examples. However, when it comes to classes with only a limited number of examples, the estimations of embedding centroids can become notably biased. In datasets featuring a large number of classes like ImageNet1k (Deng et al. (2009)), certain categories might remain unassigned or have very few examples assigned to them until the adaptation process concludes. This situation reduces the reliability of centroid estimation for these classes. Consequently, it becomes imperative to implement a mechanism for filtering out predictions with low reliability.

On the other hand, we follow TPT (Manli et al. (2022)) to leverage random augmentations to improve the prediction quality on test examples. For each test example $x_i$, we prepare $B$ augmented views $\{x_i^1, ..., x_i^B\}$, which result in a $B$ distinct predictions $\{p_i^1, ..., p_i^B\}$ that also requires to be filtered and to preserve the reliable ones. As described in Equation 1, TPT selects the predictions $p_i^b$ by thresholding their entropy $H(p_i^b) > \tau$, as the high entropy predictions tend to be more confident.

On the contrary, we draw inspiration from a study from the area of speech recognition Laptev & Ginsburg (2023) and opt for Rényi Entropy to estimate the reliability of each prediction. This decision is motivated by the observed stronger correlation between Rényi Entropy and prediction accuracy, as indicated in the study. For each test example $x_i$, we generate regular predictions $p_i^b$ by calculating the softmax-cosine similarity between visual embedding $v_i^b$ and text embedding $t_j$:

$$p_i^b = softmax\left(\left[cos(v_i^b, t_j)\right]_{j=1}^J\right), \qquad\qquad (4)$$

---

**Algorithm 1** BaFTA: Backprop-Free Test-Time Adaptation for zero-shot VLM.

---

**Require:** Vision Language Pre-trained Model $M = \{M_v, M_t\}$
**Require:** Test Samples $X = \{x_i\}_{i=1}^I$; Class Names $C = \{c_j\}_{j=1}^J$; Template Prompts $\{\theta_z\}_{z=1}^Z$

$\quad t_j \leftarrow \frac{1}{Z} \sum_z M_t(\theta_z(c_j))$ $\qquad\qquad$ ▷ Prepare multi-templates text embeddings for each class
$\quad \hat{t}_j \leftarrow P^*(t_j | \{t_1, ..., t_J\})$ $\qquad\qquad\qquad\qquad$ ▷ Projected text embeddings (Eq 3)
$\quad w_j \leftarrow \hat{t}_j, k_j \leftarrow 0$ $\qquad\qquad$ ▷ Initialize class centroids $w_j$ and counter $k_j$ for each class
$\quad$**for** i $\leftarrow$ 1 to $I$ **do**
$\qquad \{x_i^b\}_{b=1}^B \leftarrow A(x_i)$ $\qquad\qquad$ ▷ Generate $B$ views with random augmentation function $A(\cdot)$
$\qquad v_i^b \leftarrow M_v(x_i^b)$ $\qquad\qquad\qquad\qquad$ ▷ Visual embedding for each augmented views
$\qquad \hat{v_i^b} \leftarrow P^*(v_i^b)$ $\qquad\qquad\qquad\qquad$ ▷ Projected visual embedding (Eq 3)
$\qquad p_i^b \leftarrow softmax\left(\left[cos(v_i^b, t_j)\right]_{j=1}^J\right)$
$\qquad\qquad\qquad$ ▷ Cosine-similarity between visual embedding $v_i^b$ and text embedding $t_j^b$, (Eq 4)
$\qquad \hat{p}_i^b \leftarrow softmax\left(\left[cos(\hat{v_i^b}, w_j)\right]_{j=1}^J\right)$
$\qquad\qquad$ ▷ Cosine-similarity between projected visual embedding $\hat{v_i^b}$ and class centroids $w_j^b$, (Eq 5)
$\qquad \tilde{p}_i \leftarrow \frac{1}{R} \sum_b Re(p_i^b) p_i^b + \frac{1}{R} \sum_b Re(\hat{p}_i^b) \hat{p}_i^b$ $\qquad$ ▷ Prediction Aggregation (Eq. 6)
$\qquad y_i \leftarrow \arg\max_j \tilde{p}_i$ $\qquad\qquad\qquad$ ▷ Get prediction for example $x_i$
$\qquad \hat{v}_i \leftarrow \frac{1}{B} \sum_{b=1}^B \hat{v_i^b}$
$\qquad w_j \leftarrow (k_j w_j + \hat{v}_i) / \|(k_j w_j + \hat{v}_i)\|, k_j \leftarrow k_j + 1$ for $j = y_i$
$\qquad\qquad\qquad$ ▷ Updates centroids and counter on predicted class $y_i$ (Eq. 2)
$\qquad$Output $y_i$ as prediction for $x_i$
$\quad$**end for**

---

and also online-clustering predictions $p_i^b$ by comparing $v_i^b$ with the class centroids $w_j$:

$$\hat{p}_i^b = softmax\left(\left[cos(P^*(v_i^b), w_j)\right]_{j=1}^J\right). \tag{5}$$

Note that we use projected visual embeddings $P^*(v_i^b)$ to calculate $\hat{p}_i^b$, because $w_j$ are calculated in the projection space. Then, we estimate the reliability of each prediction $p$ with the Rényi entropy:

$$Re(p) = \frac{1}{\alpha - 1} \log \sum_{j=1}^J (p[j])^\alpha$$

Finally, we aggregate the predictions $\{p_i^b\}$ and $\{\hat{p}_i^b\}$ with their Rényi entropy as the weight:

$$\tilde{p}_i = \frac{1}{R}\left(\sum_{b=1}^B Re(p_i^b) p_i^b + \sum_{b=1}^B Re(\hat{p}_i^b) \hat{p}_i^b\right)$$
$$= \frac{1}{R}(p_i + \hat{p}_i) \tag{6}$$

where $R = \sum_{b=1}^B (Re(p_i^b) + Re(\hat{p}_i^b))$ is the normalization factor to ensure $\tilde{p}_i$ sums to 1.

## 3.4 ALGORITHM AND OVERVIEW

We demonstrate the overview of BaFTA in Figure 1. Instead of employing prompt-tuning, which entails back-propagation and the risk of potential model collapse during unsupervised training, BaFTA takes a backpropagation-free approach. We directly refine the class embeddings with online clustering (as detailed in Section 3.1) in a projection space that aligns the visual and text embeddings (as detailed in Section 3.2). For each test instance, BaFTA generates two sets of predictions. The first set follows the standard contrastive VLM classification protocol, measuring cosine similarity between visual embeddings of augmented views and the text embeddings of class names. The second set measures cosine similarity between visual embeddings and centroids obtained through online clustering. These predictions are subsequently combined, considering their reliability as evaluated by Rényi Entropy (as outlined in Section 3.3), to yield the final results. For a comprehensive understanding of BaFTA's procedures, please also refer to Algorithm 1.

| | ImageNet | ImageNet-A | ImageNet-V2 | ImageNet-R | ImageNet-Sketch | NDS Avg |
|---|---|---|---|---|---|---|
| CLIP (ViT-B/16) | 66.73 | 47.87 | 60.86 | 73.98 | 46.09 | 57.20 |
| Multi-Template | 68.34 | 49.89 | 61.88 | 77.65 | 48.24 | 59.42 |
| Hierarchy-CLIP | 68.86 | 31.07 | 62.00 | 60.62 | 48.26 | 50.48 |
| TPT | 68.98 | 54.77 | 63.45 | 77.06 | 47.94 | 60.81 |
| BaFTA | **71.43** | **58.19** | **64.46** | **79.06** | **50.51** | **63.06** |
| CoOp (16-shot) | 71.51 | 49.71 | 64.20 | 75.21 | 47.99 | 59.28 |
| TPT + CoOp | 73.61 | 57.95 | 66.83 | 77.27 | 49.29 | 62.84 |
| BaFTA + CoOp | **74.42** | **59.21** | **67.15** | **79.00** | **51.39** | **64.19** |
| CLIP (RN50) | 58.16 | 21.83 | 51.41 | 56.15 | 33.37 | 40.69 |
| Multi-Template | 59.81 | 23.24 | 52.91 | **60.72** | 35.48 | 43.09 |
| TPT | 60.74 | 26.67 | 54.70 | 59.11 | 35.09 | 43.89 |
| BaFTA | **62.01** | **26.91** | **55.26** | 59.79 | **36.37** | **44.58** |
| CoOp (16-shot) | 63.33 | 23.06 | 55.40 | 56.60 | 34.67 | 42.43 |
| TPT + CoOp | 64.73 | **30.32** | 57.83 | 58.99 | 35.86 | 45.75 |
| BaFTA + CoOp | **65.92** | 29.39 | **58.22** | **59.45** | **36.84** | **45.98** |

Table 3: Comparison of top-1 accuracy on ImageNet and the Natural Distribution Shifts (NDS) Benchmarks. All methods evaluated in zero-shot classification setting, except CoOp being fine-tuned on ImageNet with 16 examples per category.

| | Average | Cars | Caltech101 | DTD | EuroSAT | FGVC | Food101 | Flower102 | Pets | UCF101 | SUN397 |
|---|---|---|---|---|---|---|---|---|---|---|---|
| CLIP (ViT B/16) | 63.58 | 65.48 | 93.35 | 44.27 | 42.01 | 23.67 | 83.65 | 67.44 | 88.25 | 65.13 | 62.59 |
| Multi-Template | 64.59 | 66.11 | 93.55 | 45.04 | 50.42 | 23.22 | 82.86 | 66.99 | 86.92 | 65.16 | 65.63 |
| CoOp (16-shot) | 63.88 | 64.51 | 93.70 | 41.92 | 46.39 | 18.47 | 85.30 | 68.71 | 89.14 | 66.55 | 64.15 |
| TPT | 65.10 | 66.87 | **94.16** | 47.75 | 42.44 | 24.78 | 84.67 | 68.98 | 87.79 | 68.04 | 65.50 |
| BaFTA | **68.52** | **69.44** | 94.08 | **50.30** | **50.49** | **27.00** | **87.03** | **73.81** | **92.61** | **71.13** | **69.34** |
| CLIP (RN50) | 55.82 | 55.70 | 85.88 | 40.37 | 23.69 | 15.66 | 73.97 | 61.75 | 83.57 | 58.84 | 58.80 |
| Multi-Template | 56.63 | 55.89 | 87.26 | 40.37 | 25.79 | 16.11 | 74.82 | 62.77 | 82.97 | 59.48 | 60.85 |
| CoOp (16-shot) | 56.18 | 55.32 | 86.53 | 37.29 | 26.20 | 15.12 | 75.59 | 61.55 | 87.00 | 59.05 | 58.15 |
| TPT | 57.66 | **58.46** | 87.02 | 40.84 | 28.33 | 17.58 | 74.88 | 62.69 | 84.49 | 60.82 | 61.46 |
| BaFTA | **63.20** | 58.29 | **87.95** | **44.03** | **39.26** | **18.15** | **77.69** | **66.67** | **88.76** | **64.26** | **62.99** |

Table 4: Top-1 Accuracy on 10 Fine-grained Benchmarks. All baselines are evaluated in zero-shot classification setting, except CoOp being fine-tuned on ImageNet with 16 examples per category.

# 4 EXPERIMENT AND RESULTS

**Baselines.** We conduct experiments in comparison of BaFTA with several benchmark models and algorithms. Our comparisons include the baseline model CLIP (Radford et al. (2021)) and the state-of-the-art test-time prompt-tuning algorithm TPT (Manli et al. (2022)) that were introduced in Section 2. For CLIP, we report both single template (denoted as CLIP), and multi-template versions. We also include Hierarchy-CLIP (Ge et al. (2023)) in the ImageNet evaluation, as it enhances prompt quality with a training-free method that leverages the ImageNet class hierarchy. Furthermore, we have introduced CoOp (Zhou et al. (2022b)), a few-shot prompt-tuning method, as an additional baseline model for both comparison and adaptation, aligning with experiments from TPT.

**Datasets.** We have conducted our experiments over two sets of datasets, following the experiment setup of Manli et al. (2022) and Zhou et al. (2022b), which includes: **1)** ImageNet Robustness Evaluation with ImageNet (Deng et al. (2009)) and its Natural Distribution Shift (NDS) variants ImageNet-V2 (Recht et al. (2019)), ImageNet-R (Hendrycks et al. (2021a)), ImageNet-Sketch (Wang et al. (2019)) and ImageNet-A (Hendrycks et al. (2021b)); **2)** Fine-Grained Datasets with Stanford Cars (Krause et al. (2013)), Caltech101 (Li et al. (2022)), Describable Textures (DTD, Cimpoi et al. (2014)), EuroSAT (Helber et al. (2019)), FGVC Aircrafts (Maji et al. (2013)), Food101 (Bossard et al. (2014)), Flowers102 (Nilsback & Zisserman (2008)), Oxford-IIIT-Pets (Parkhi et al. (2012)), UCF101 (Soomro et al. (2012)) and SUN397 (Xiao et al. (2010)).

| | Average | ImageNet | ImageNet-A | ImageNet-V2 | ImageNet-R | ImageNet-S | Cars | Caltech101 | DTD | EuroSAT | FGVC | Food101 | Flower102 | Pets | UCF101 | SUN397 |
|---|---|---|---|---|---|---|---|---|---|---|---|---|---|---|---|---|
| CLIP | 63.46 | 68.34 | 49.89 | 61.88 | 77.65 | 48.24 | 66.11 | 93.55 | 45.04 | 50.42 | 23.22 | 82.86 | 66.99 | 86.92 | 65.16 | 65.63 |
| BaFTA-RA | 65.88 | 70.53 | 57.87 | 64.45 | 79.03 | 49.40 | 67.96 | 93.87 | 46.93 | 47.88 | **27.09** | 86.66 | 71.42 | 89.23 | 69.07 | 66.74 |
| BaFTA-OC | 62.53 | 66.77 | 52.43 | 50.15 | 74.74 | 46.65 | 64.99 | 92.12 | 49.41 | 49.52 | 24.78 | 85.72 | 58.60 | 89.31 | 68.60 | 64.11 |
| BaFTA | **67.26** | **71.43** | **58.19** | **64.46** | **79.06** | **50.51** | **69.44** | **94.08** | **50.30** | **50.49** | 27.00 | **87.03** | **73.81** | **92.61** | **71.13** | **69.34** |

Table 5: Comparison over different BaFTA predictions. BaFTA-RA (Rényi Aggregation): standard predictions over augmented views aggregated with Rényi Entropy; BaFTA-OC (Online Clustering): predictions generated with the clustering centroids; BaFTA-RA, BaFTA-OC, BaFTA refers to the $p_i$, $\hat{p}_i$ and $\tilde{p}_i$ from Eq. 6 respectively. All results produced with CLIP (ViT-B/16).

| | Average | ImageNet | ImageNet-A | ImageNet-V2 | ImageNet-R | ImageNet-S | Cars | Caltech101 | DTD | EuroSAT | FGVC | Food101 | Flower102 | Pets | UCF101 | SUN397 |
|---|---|---|---|---|---|---|---|---|---|---|---|---|---|---|---|---|
| $k$NN w/o $P^*$ | 61.56 | 63.03 | 44.29 | 50.56 | 71.19 | 44.29 | 62.49 | 92.29 | 43.85 | 58.37 | 22.35 | 68.66 | 84.56 | 85.01 | 68.68 | 63.77 |
| $k$NN w/ $P^*$ | 64.04 | 66.62 | 48.91 | 55.41 | 77.91 | 47.62 | 67.01 | 93.55 | 45.74 | 53.75 | 23.52 | 69.35 | 86.33 | 89.64 | 69.36 | 65.87 |

Table 6: Effectiveness of Projection $P^*$ (Eq. 3) in improving embedding distribution. Results produced with CLIP (ViT-B/16) embeddings, demonstrated by the top-1 accuracy improvement of $k$NN classifier with $k = 5$. Columns correspond to the columns in Table 5.

**Implementation Details.** In our experiments, we employ the ViT-B/16 and ResNet50 checkpoints from CLIP as the baseline models for comparison and adaptation. In line with the TPT implementation, we utilize a simple combination of `RandomResizedCrop` and `RandomFlip` to prepare 63 augmented views, constituting a mini-batch of 64 images for each test image. This choice, as previously observed in Manli et al. (2022), strikes a suitable balance between runtime efficiency and performance. We have employed the exponential form of Renyi Entropy with order $\alpha = 0.5$ following Laptev & Ginsburg (2023). For experiments on-top-of the CoOp, we use the 16-shot fine-tuned model and ensemble the predictions generated from CoOp embeddings with our predictions using Rényi entropy. Instead of directly replacing the prompts, we adopt this approach because we have observed that CoOp embeddings sometimes perform less effectively than the multi-template embeddings provided by CLIP. For all other BaFTA results, we use official template sets provided by CLIP to generate the text embeddings. Unless otherwise specified, all BaFTA results are reported with a warm-up schedule of $10J$ examples ($J$ as number of class) before the online clustering predictions aggregated into final prediction. For the embedding projection matrix, we use $U' = [e_2, ..., e_J]$ for all datasets, except for datasets with more than 150 categories such as ImageNet, we use $U' = [e_2, ..., e_{150}]$ for best performance.

## 4.1 MAIN RESULTS

In Table 3 and Table 4 we present the comprehensive results of backpropagation-free test-time algorithm BaFTA in comparison to baseline methods across five ImageNet robustness benchmarks and ten fine-grained classification benchmarks.

As illustrated in Table 3, BaFTA exhibits a substantial improvement over the baseline CLIP model on ImageNet, achieving enhancements of 3.07% and 6.11% on ViT-B/16 and RN50 models, respectively. Notably, BaFTA achieves these results without the need for backpropagation training during adaptation. Furthermore, BaFTA surpasses the state-of-the-art test-time prompt tuning method TPT by notable margins of 2.45% and 1.17% on ViT-B/16 and RN50. Additionally, when applied on top of the few-shot fine-tuned prompts from CoOp, BaFTA further enhances CoOp's performance by significant margins. On the Natural Distribution Shifts Benchmarks, BaFTA even outperforms CoOp with a remarkable margin of 3.78% and 2.15% on the ViT-B/16 and RN50 models (as well as on the fine-grained datasets as shown in Table 4). This indicates that test-time adaptation provides superior results compared to cross-domain generalization through few-shot supervised methods. In Table 4, BaFTA exhibits even larger improvements over TPT on the fine-grained datasets, with notable margins of 3.42% and 5.54% on ViT-B/16 and RN50, respectively. This performance improvement is possibly attributed to the better online clustering performance on datasets with fewer target categories. These results underline the effectiveness of BaFTA, even without the use of backpropagation training, solidifying its position as a valuable and robust test-time adaptation method.

| CLIP | $\overline{p^b}$ | $\overline{\text{softmax}(p^b)}$ | $\overline{\max(p^b)p^b}$ | $\overline{\mathbb{1}[H(p^b) < \tau]p^b}$ | $\overline{\hat{H}(p^b)p^b}$ | $\overline{Re(p^b)p^b}$ | | |
|---|---|---|---|---|---|---|---|---|
| | | | | | | $\alpha = 0.25$ | $\alpha = 0.5$ | $\alpha = 0.75$ |
| 68.34 | 69.43 | 70.19 | 58.69 | 70.40 | 69.87 | 70.52 | **70.53** | 70.30 |

Table 7: Comparison of different methods to aggregate predictions $p^b$ from augmented views. All results are top-1 accuracy reported with CLIP (ViT-B/16) on ImageNet with 64 augmented views for each test example. Please refer to the text for details on the notation.

## 4.2 ABLATION STUDIES

**Comparison on Predictions from BaFTA.** Table 5 presents the ablation results of BaFTA, assessing the accuracy of each of its prediction sets: $\hat{p}_i$, $p_i$, and $\tilde{p}_i$, as described in Equation 6. The results reveal that simply applying Rényi Entropy to aggregate predictions from augmented views results in a 2.40% average accuracy improvement across the 15 datasets. The predictions generated with the online clustering centroids improve CLIP's performance on ImageNet-A, DTD, FGVC, Food101, Oxford-Pets, and UCF101, but do not show improvement on others. This discrepancy may be attributed to two factors: 1) The accuracy is calculated over the entire dataset, where the clustering centroids are not yet stable on earlier examples; 2) Some of the clustering centroids might become unreliable, on datasets with biased distribution, or a large number of categories, such as ImageNet and SUN397. However, thanks to Rényi Entropy Aggregation, BaFTA is capable of leveraging the reliable predictions among the clustering-based predictions ($\hat{p}_i$) and achieves an additional 1.40% improvement over $p_i$, resulting in a total improvement of 3.80% across the 15 datasets.

**Effectiveness of Projected Embedding Space.** Table 6 provides evidence of the effectiveness of the Projection $P^*$ in enhancing the distribution of CLIP embeddings for clustering, as proposed in Hu et al. (2023). The results demonstrate a 2.48% improvement in averaged $k$-nearest neighbor (kNN) classifier accuracy across the 15 datasets after projecting the CLIP embeddings with $P^*$. This improvement signifies that $P^*$ successfully enhances the neighboring relationships among CLIP embeddings in the projection space, which, in turn, will benefit the online clustering process.

**Comparison on Prediction Aggregation Method.** In Table 7, we present the ablation results on choice of aggregation function that merges the predictions results from augmented views. We use the over-line $\overline{X} = \sum_{b=1}^{B} X_b$ to denote the average over $B$ augmented views. From left to right, we have: 1) CLIP: baseline prediction without augmentation; 2) $\overline{p^b}$: averaged prediction; 2) $\overline{\text{softmax}(\gamma p^b)}$: soft majority-vote prediction; 3) $\overline{\max(p^b)p^b}$: weighted-average prediction with confidence estimated by maximum entry of $p^b$; 4) $\overline{\mathbb{1}[H(p^b) > \tau]p^b}$: average of low-entropy (high-confidence) predictions, as adopted by TPT; 6) $\hat{H}(p^b)p^b$: weighted-average prediction with confidence estimated by the normalized entropy $\hat{H}(p^b) = (H^{max} - H(p^b))/H^{max}$; 6) $\overline{Re(p^b)p^b}$: weighted-average prediction with confidence estimated by Rényi entropy $Re(p^b)$, with entropy order $\alpha = 0.25, 0.50, 0.75$. As shown in the Table, Rényi entropy at order of 0.50 provides the best results over all the other options.

## 5 CONCLUSION

In this work, we have focused on enhancing the performance of large-scale pre-trained vision-language models, exemplified by CLIP, in the context of zero-shot image classification. While various test-time prompt tuning methods have been developed to refine class embeddings during inference, they often grapple with the challenge of selecting appropriate learning rates in the absence of validation data during test-time training. To address this challenge, we have introduced a novel backpropagation-free method for test-time adaptation in vision-language models. Instead of fine-tuning text prompts to refine class embeddings, our approach directly estimates class centroids using online clustering within a projected embedding space that aligns text and visual embeddings. We have also proposed a dynamic aggregation technique for predictions, leveraging both estimated and original class embeddings, as well as distinct augmented views. This aggregation is guided by an assessment of prediction reliability using Rényi entropy. Our comprehensive experimentation has consistently demonstrated that our approach outperforms state-of-the-art test-time adaptation methods by a significant margin. This work contributes to improving vision-language models, offering a practical solution for real-world applications.

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
