# SUPPLEMENTARY MATERIALS FOR
# BaFTA: BACKPROP-FREE TEST-TIME ADAPTATION FOR ZERO-SHOT VISUAL LANGUAGE MODELS

## APPENDIX A: INFERENCE TIME EFFICIENCY ANALYSIS

| Time Per Example (ms) | TPT | BaFTA |
|---|---|---|
| RN50 | 841.0 | 158.7 |
| ViT-B/16 | 873.0 | 183.8 |

Table 1: Inference time comparison of TPT and BaFTA. All results reported with ImageNet examples, evaluated on NVIDIA A40 GPU.

In Table 1, we present a comparison of inference times between TPT and BaFTA using both ViT-B/16 and RN50 backbones. The inference time per example (in milliseconds) was calculated by recording the total time required to complete a 10,000-iteration inference on ImageNet examples, with a single example processed per iteration. All experiments were conducted on a computation node equipped with an AMD EPYC 7313 CPU (32 cores), 256 GB memory, and a single NVIDIA A40 GPU (48GB).

As indicated in the table, BaFTA exhibits a notable advantage, being approximately 5 times faster than TPT. The significant difference in inference time for TPT can be attributed to two main factors: 1) TPT requires two forward passes and one backward pass in each iteration, whereas BaFTA requires only a single forward pass; 2) TPT requires recomputation of classification embeddings through the text encoder at each forward pass, while BaFTA conducts the text encoder once offline and updates the classification embeddings directly in the embedding space during inference.

## APPENDIX B: ABLATION STUDY ON RÉNYI ENTROPY ORDER $\alpha$

In order to assess the sensitivity of Rényi Entropy aggregation performance to the entropy order $\alpha$, we analyze the accuracy of Rényi Entropy aggregated predictions from augmented views with varying $\alpha$, specifically $\alpha \in \{0.1, 0.2, 0.3, 0.4, 0.5, 0.6, 0.7, 0.8, 0.9, 0.99\}$, over all 15 datasets used in our study. In all experiments, we employ CLIP-ViT-B/16 as the base model and evaluate BaFTA-RA to investigate the influence of $\alpha$ on prediction aggregation without the impact of online clustering results.

Figure 1 illustrates the $\alpha-$accuracy curve across all 15 datasets. The curves are normalized by subtracting the maximum value within each curve, ensuring they are plotted within the same value range. The bold red curves represent the averaged accuracy over the 15 datasets, revealing that the average performance peaks at $\alpha = 0.5$ and $\alpha = 0.6$. Additionally, the plot indicates that prediction aggregation accuracy is relatively insensitive to the choice of $\alpha$, with most curves exhibiting less than a 0.3% change in accuracy across the $\alpha$ range [0.1, 0.99]. Most datasets achieve peak performance with $\alpha$ in the range of [0.3, 0.8], and selecting $\alpha = 0.5$ guarantees the performance to be within 0.25% from the peak.

In Table 2, we display the accuracy standard deviation of Rényi Entropy aggregated predictions over varying $\alpha$. The table reveals that the accuracy standard deviation is less than 0.3% on most datasets, with the exception of ImageNet-A (corresponding to the orange curve in Figure 1). ImageNet-A is composed of challenging outlier examples where machine learning models often falter. It is

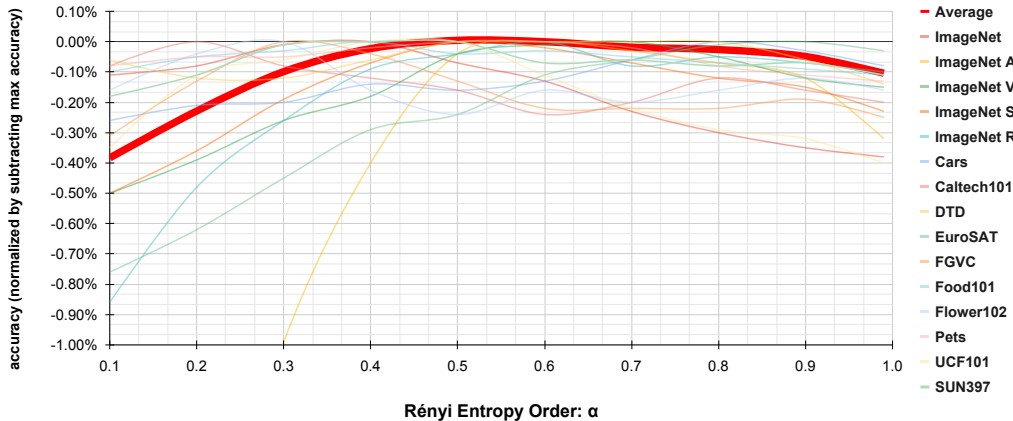

Figure 1: $\alpha$-accuracy curves on 15 datasets, with $\alpha \in [0.1, 0.99]$. In order to fit all curves into one plot with unified value range, all curves are normalized by subtracting the maximum accuracy within the curve. The bold red curve represents the averaged accuracy over 15 datasets, achieves its maximum value at $\alpha = 0.5$ and $\alpha = 0.6$. This plot indicates that prediction aggregation accuracy is not highly sensitive to the choice of $\alpha$, with most curves exhibiting less than a 0.3% change in accuracy across the $\alpha$ range [0.1, 0.99]

| | Average | ImageNet | ImageNet-A | ImageNet-V2 | ImageNet-R | ImageNet-S | Cars | Caltech101 | DTD | EuroSAT | FGVC | Food101 | Flower102 | Pets | UCF101 | SUN397 |
|---|---|---|---|---|---|---|---|---|---|---|---|---|---|---|---|---|
| accuracy std. (%) | 0.12 | 0.14 | 0.93 | 0.17 | 0.16 | 0.27 | 0.08 | 0.07 | 0.06 | 0.27 | 0.10 | 0.03 | 0.07 | 0.05 | 0.15 | 0.06 |

Table 2: Accuracy standard deviation of Rényi Entropy aggregated predictions over varying $\alpha$.

possible that CLIP produces less confident and flatter prediction logits on ImageNet-A, rendering its performance more sensitive to variations in $\alpha$ compared to other datasets.

## APPENDIX C: MORE ABLATION STUDIES

In additional to the ablation results we presented in Table 5 of the main paper, we present two more versions of BaFTA for ablation studies on templates and Rényi Entropy aggregation.

### C.1 SINGLE AND MULTIPLE TEMPLATES

In Table 3, we introduce results of BaFTA-*single* for ablation studies on the templates. BaFTA-*single* employs the single-template CLIP as the base model for adaptation, resulting in a notable 6.22% improvement in averaged accuracy across 15 datasets. Although BaFTA-*single* exhibits slightly less improvement compared to TPT, it boasts the advantages of no back-propagation requirement and five times faster inference speed. Furthermore, BaFTA is capable of taking advantage from the multi-template ensemble (as suggested by official CLIP), achieving an additional 4.12% improvement and achieves 67.26% averaged acuracy on 15 datasets, which is not attainable by TPT.

### C.2 EFFECTIVENESS OF RÉNYI ENTROPY AGGREGATION

In Table 3, we introduce results of BaFTA-Avg for additional ablation studies on the effectiveness of Rényi Entropy aggregation. In the BaFTA-Avg experiment, we substitute the Rényi Entropy aggregation with a simple average function to merge predictions from online clustering and augmentations. Results demonstrate that BaFTA-Avg yields a 2.12% improvement over the base model

| | Average | ImageNet | ImageNet-A | ImageNet-V2 | ImageNet-R | ImageNet-S | Cars | Caltech101 | DTD | EuroSAT | FGVC | Food101 | Flower102 | Pets | UCF101 | SUN397 |
|---|---|---|---|---|---|---|---|---|---|---|---|---|---|---|---|---|
| CLIP *single* | 56.92 | 66.73 | 47.87 | 60.86 | 73.98 | 46.09 | 55.70 | 85.88 | 40.37 | 23.69 | 15.66 | 73.97 | 61.75 | 83.57 | 58.84 | 58.80 |
| TPT | 64.21 | 68.98 | 54.77 | 63.45 | 77.06 | 47.94 | 66.87 | **94.16** | 47.75 | 42.44 | 24.78 | 84.67 | 68.98 | 87.79 | 68.04 | 65.50 |
| BaFTA *single* | 63.14 | 66.31 | 55.73 | 61.17 | 76.00 | 45.89 | 66.78 | 91.12 | 45.92 | 40.95 | 24.30 | 85.92 | 66.54 | 85.91 | 67.99 | 66.55 |
| CLIP *multi* | 63.46 | 68.34 | 49.89 | 61.88 | 77.65 | 48.24 | 66.11 | 93.55 | 45.04 | 50.42 | 23.22 | 82.86 | 66.99 | 86.92 | 65.16 | 65.63 |
| BaFTA-RA | 65.88 | 70.53 | 57.87 | 64.45 | 79.03 | 49.40 | 67.96 | 93.87 | 46.93 | 47.88 | **27.09** | 86.66 | 71.42 | 89.23 | 69.07 | 66.74 |
| BaFTA-OC | 62.53 | 66.77 | 52.43 | 50.15 | 74.74 | 46.65 | 64.99 | 92.12 | 49.41 | 49.52 | 24.78 | 85.72 | 58.60 | 89.31 | 68.60 | 64.11 |
| BaFTA-Avg | 65.58 | 69.40 | 53.55 | 63.22 | 76.56 | 48.57 | 67.97 | 93.61 | 48.97 | 48.43 | 26.28 | 85.96 | 72.77 | 91.18 | 69.27 | 67.94 |
| BaFTA | **67.26** | **71.43** | **58.19** | **64.46** | **79.06** | **50.51** | **69.44** | 94.08 | **50.30** | **50.49** | 27.00 | **87.03** | **73.81** | **92.61** | **71.13** | **69.34** |

Table 3: Additional ablation studies on BaFTA components. All results produced with CLIP (ViT-B/16). In addition to the versions presented in Table 5 from the main paper (BaFTA-RA, BaFTA-OC, BaFTA), we include additional rows for BaFTA-*single* and BaFTA-Avg. BaFTA-*single* utilizes the single-template CLIP as the base model, while BaFTA-Avg combines online clustering and augmentation predictions with a naive average instead of weighting with Rényi Entropy.

| | ImageNet | ImageNet-A | ImageNet-V2 | ImageNet-R | ImageNet-Sketch | NDS Avg |
|---|---|---|---|---|---|---|
| CLIP (ViT-B/16) | 66.73 | 47.87 | 60.86 | 73.98 | 46.09 | 57.20 |
| Multi-Template | 68.34 | 49.89 | 61.88 | 77.65 | 48.24 | 59.42 |
| Hierarchy-CLIP | 68.86 | 31.07 | 62.00 | 60.62 | 48.26 | 50.48 |
| TPT | 68.98 | 54.77 | 63.45 | 77.06 | 47.94 | 60.81 |
| BaFTA | **71.43** | **58.19** | **64.46** | **79.06** | **50.51** | **63.06** |
| CoOp (16-shot) | 71.51 | 49.71 | 64.20 | 75.21 | 47.99 | 59.28 |
| CoCoOp (16-shot) | 71.02 | 50.63 | 64.07 | 76.18 | 48.75 | 59.91 |
| ProGrad (4-shot) | 70.45 | 49.45 | 63.35 | 75.21 | 48.17 | 59.05 |
| TPT+CoOp | 73.61 | 57.95 | 66.83 | 77.27 | 49.29 | 62.84 |
| TPT+CoCoOp | 71.07 | 58.47 | 64.85 | 78.65 | 48.47 | 62.61 |
| BaFTA + CoOp | **74.42** | **59.21** | **67.15** | **79.00** | **51.39** | **64.19** |
| CLIP (RN50) | 58.16 | 21.83 | 51.41 | 56.15 | 33.37 | 40.69 |
| Multi-Template | 59.81 | 23.24 | 52.91 | 60.72 | 35.48 | 43.09 |
| TPT | 60.74 | 26.67 | 54.70 | 59.11 | 35.09 | 43.89 |
| SuS-X (Stable Diff) | 61.84 | - | - | 61.76 | 36.30 | - |
| SuS-X (LAION-5B) | 61.89 | - | - | **62.10** | **37.83** | - |
| BaFTA | **62.01** | **26.91** | **55.26** | 59.79 | 36.37 | **44.58** |
| CoOp (16-shot) | 63.33 | 23.06 | 55.40 | 56.60 | 34.67 | 42.43 |
| CoCoOp (16-shot) | 62.81 | 23.32 | 55.72 | 57.74 | 34.48 | 42.82 |
| ProGrad (4-shot) | 62.17 | 23.05 | 54.79 | 56.77 | 34.40 | 42.25 |
| TPT+CoOp | 64.73 | **30.32** | 57.83 | 58.99 | 35.86 | 45.75 |
| TPT+CoCoOp | 62.93 | 27.40 | 56.60 | **59.88** | 35.43 | 44.83 |
| BaFTA + CoOp | **65.92** | 29.39 | **58.22** | 59.45 | **36.84** | **45.98** |

Table 4: Comparison of top-1 accuracy on ImageNet and the Natural Distribution Shifts (NDS) Benchmarks. All methods evaluated in zero-shot classification setting, except 1) CoOp, CoCoOp and ProGrad are fine-tuned on ImageNet with 4 or 16 examples per category; 2) SuS-X uses external knowledge such as Stable Diffusion or LAION-5B to construct support set.

CLIP-*multi*, highlighting the effectiveness of augmentations and online clustering. Furthermore, BaFTA achieves an additional 1.68% improvement over BaFTA-Avg, underscoring the effectiveness of Rényi Entropy aggregation over naive average assembly.

## APPENDIX D: IMAGENET RESULTS WITH ADDITIONAL BASELINES

In Table 4, we expanded upon Table 3 from the main paper to include additional baselines, namely CoCoOp (Zhou et al. (2022)), SuS-X (Udandarao et al. (2022)), and ProGrad (Zhu et al. (2023)).

The performance of CoCoOp and ProGrad are similar CoOp on both RN50 and ViT-B/16. BaFTA achieves comparable performance to ProGrad, CoOp, and CoCoOp on ImageNet and demonstrates superior performance on datasets with natural distribution shifts. Notably, ProGrad, CoCoOp, and

CoOp employ prompts fine-tuned with few-shot examples from ImageNet, whereas BaFTA operates without any supervision, highlighting its effectiveness and flexibility. Additionally, BaFTA can be combined with such fine-tuned prompts to provide further improvements, which significantly outperforms the baseline methods ProGrad, CoOp, CoCoOp, TPT+CoOp and TPT+CoCoOp.

BaFTA outperforms SuS-X on ImageNet, although with slightly lower performance on ImageNet-R and ImageNet-Sketch. It's noteworthy that SuS-X leverages external resources such as Stable Diffusion (Rombach et al. (2022)) and LAION-5B (Schuhmann et al. (2022)) to construct support sets and transform the zero-shot problem into a few-shot problem. In contrast, BaFTA operates without any labels or external resources, showcasing its label-free and resource-independent nature.

## APPENDIX E: EFFECTIVENESS OF PROJECTED EMBEDDING SPACE

In Figure 2, we present t-SNE plots for Oxford-IIIT-Pets, Describable Textures, and Stanford Cars to visually showcase the distribution differences between original CLIP visual embeddings and projected visual embeddings.

As illustrated by the t-SNE plots, the projection space effectively transforms sparse clusters into denser formations, leading to improved online clustering results. Additionally, we observed that the enhancement in clustering brought by the projection is potentially correlated with the classification accuracy on the respective datasets. For instance, CLIP attains a 86.92% zero-shot accuracy on Oxford-IIIT-Pets, with the projection significantly improving its clustering quality. In contrast, CLIP achieves only 45.04% accuracy on Describable Textures, and the improvement provided by the projection over the clustering condition is relatively subtle.

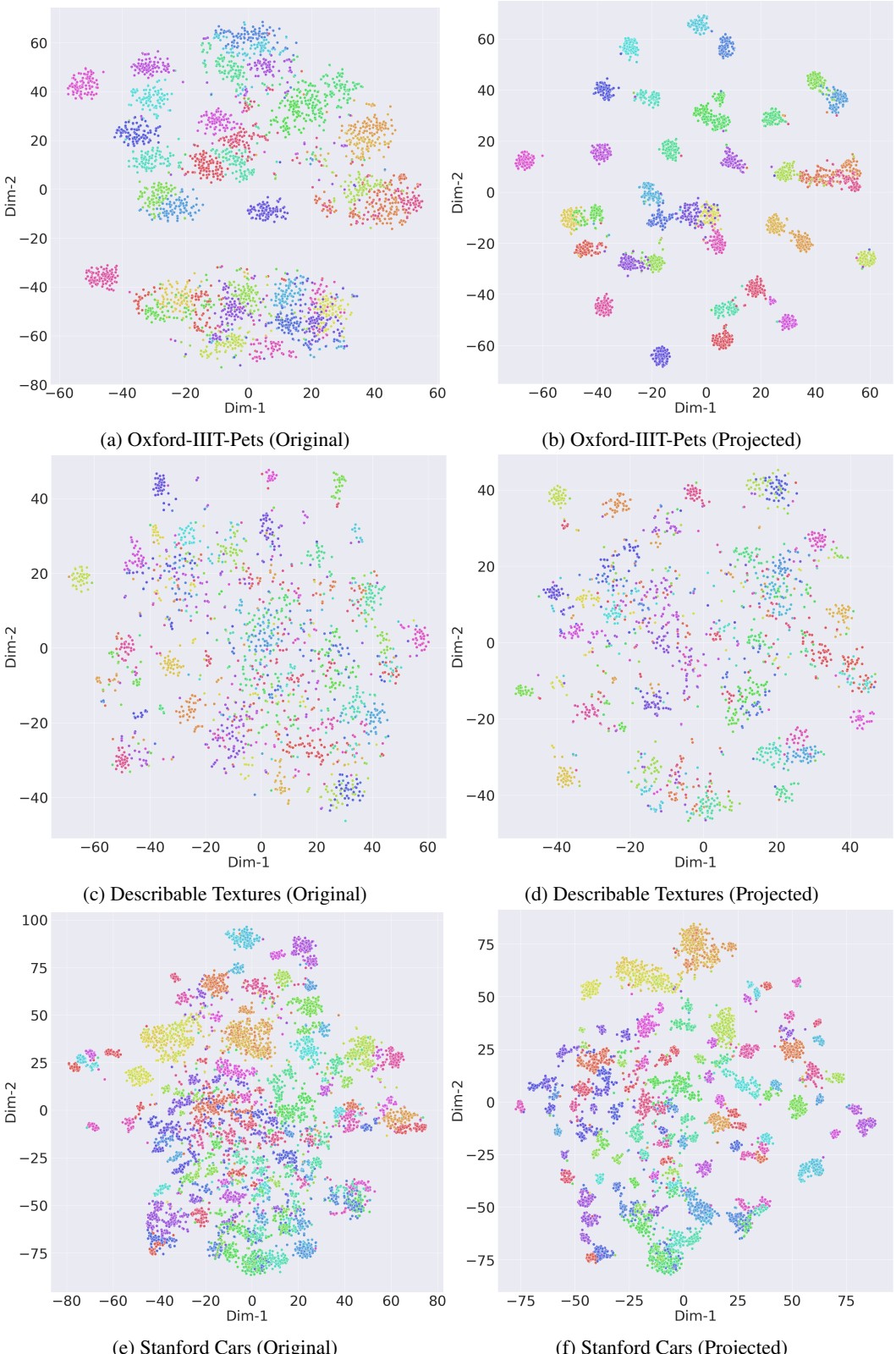

Figure 2: tSNE plots of original and projected visual embeddings from evaluation datasets.