# OpenReview forum: "BaFTA: Backprop-Free Test-Time Adaptation for Zero-shot Vision Language Models"
_ICLR.cc/2024/Conference — Submitted to ICLR 2024_

### Official Review · Reviewer_JA78 · 2023-10-28

**Soundness:** 2 fair
**Presentation:** 3 good
**Contribution:** 3 good
**Rating:** 6
**Confidence:** 3

**Summary:**

This paper proposes a new backpropagation-free method for test-time adaptation in vision-language models. Instead of fine-tuning text prompts, this paper employs online clustering within a projected embedding space that aligns text and visual embeddings to estimate class centroids. To reliably evaluate predictions, the paper utilizes Renyi entropy, which ensures accuracy and robustness. Experiments demonstrate that the method improves zero-shot classification accuracy and outperforms SOTA test-time adaptation methods.

**Strengths:**

1. This paper provides a clear introduction to the challenges of test-time prompt tuning methods and introduces the state-of-the-art method TPT. To address these issues, this paper proposes BaFTA, which is highly motivated.

2. The algorithm proposed in this paper is simple, lightweight, and its description is also very clear.

**Weaknesses:**

1. In the part of Projected Online Clustering, the main distinction between your approach and ReCLIP[1] is that your clustering is conducted online, while ReCLIP's clustering is done offline. Furthermore, due to your Test-Time adaptation setting, your clustering must be conducted online. So I think this may not represent a particularly solid innovation.

2. The experiments are insufficient. In Table 3 and Table 4, I suggest that the authors include additional methods such as CoCoOp[2], TPT+CoCoOp, BaFTA+CoCoOp, and Sus-X [3] and [4].

3. The ablation study on entropy order α should be more comprehensive. In Table 7, when entropy order equals 0.25, the result is 70.52, and when it equals 0.5, the result is 70.53. So, what about when it equals 0.4 or 0.6? Therefore, the authors should conduct more fine-grained experiments.

[1] ReCLIP: Refine Contrastive Language Image Pre-Training with Source Free Domain Adaptation. arXiv 2023.

[2] Conditional Prompt Learning for Vision-Language Models. CVPR 2022.

[3] SuS-X: Training-Free Name-Only Transfer of Vision-Language Models. CVPR 2023.

[4] Prompt-aligned Gradient for Prompt Tuning. ICCV 2023.

**Questions:**

1. The method proposed in this paper has persistence? In other words, can I save the results at a certain moment for later updates?

2. In Table 6, the authors discuss the effectiveness of projection P*. Is there any explanation regarding how projection P* can improve the embedding distribution?

**Details Of Ethics Concerns:**

No concerns.

---

> ### Author Response · Authors · 2023-11-18
>
> We appreciate the valuable feedback! Here are our responses to your concerns:
> 1. ReCLIP focuses on source-free adaptation, employing a label propagation algorithm for pseudo label generation. However, this method relies on a substantial number of examples to yield satisfactory results, a condition that may not be met in many practical scenarios. In our exploration of alternatives to the label propagation approach, we found that online clustering performs satisfactorily. This approach alleviates the need for an extensive pool of available test data while maintaining performance standards to a reasonable degree.
> 2. In response to your suggestion, we have incorporated additional methods (CoCoOp, CoCoOp+TPT, ProGrad, SuS-X) into the supplementary materials. However, we opted not to include CoCoOp+BaFTA for two reasons: 1) The authors of CoOp/CoCoOp have not released the weights for CoCoOp, preventing us from running BaFTA+CoCoOp; 2) CoOp+TPT outperforms CoCoOp+TPT on both RN50 and ViT-B/16, and we believe that comparing with the stronger CoOp+TPT is more informative. For a comprehensive examination of results and analysis, kindly refer to Section D and Table 4. We are planning to reorganize and integrate some of these methods into the main paper in future revisions.
> 3. To offer more insights into the sensitivity of the Renyi Entropy parameter, we have performed extensive experiments with detailed results in the supplementary materials. The analysis suggests that BaFTA's performance is not highly sensitive to the choice of $\alpha$. Please refer to Section B, Figure 1 and Table 2 for a more detailed analysis.
> 4. Yes, BaFTA can be easily saved at any moment. The only evolving parameters are the moving averaged centroids and the cluster member counts, making them space-efficient for storage.
> 5. The projection can be conceptually described in two steps: 1) SVD extracts the orthonormal basis U of the text embedding subspace. The projection matrix $P=UU^T$ projects everything onto the text embedding subspace, removing dimensions not participating in the classification calculation, since they are orthogonal to the text embeddings. This step removes the category agnostic information from visual embeddings and makes them easier to cluster; 2) $P^*=U’U’^T$ discards the first axis from U where all text embeddings overlap the most. This step removes shared information between text embeddings from different classes, emphasizing their differences and separating clusters from different classes.

---

### Official Review · Reviewer_sCvw · 2023-10-30

**Soundness:** 3 good
**Presentation:** 3 good
**Contribution:** 2 fair
**Rating:** 5
**Confidence:** 3

**Summary:**

This paper proposes to use centroids of visual embedding to improve the test time performance of CLIP. The proposed method is training-free.

**Strengths:**

- 1. It is interesting to see such a training-free method achieves good performance compared to previous tuning methods like TPT.

- 2. The proposed method is training-free. The authors managed to combine different components to make them work well.

**Weaknesses:**

- 1. A more comprehensive ablation of different components in the system is needed. In algorithm 1, the authors use $Eq.3$, online clustering, weighted aggregation, etc. I am confused about the contributions of different components. For example, in Table 5, the authors can add results of a simple average of BaFTA-OC + CLIP text predictions. This may help us understand the effectiveness of Renyi Aggregation more clearly. Other detailed ablations are also needed.

- 2. The core idea of the proposed method is to use visual centroids to help classifications. The online updating paradigm also assumes the test data come from the same distribution. In practice, this may not be the case. For instance, test samples may come from different distributions. One possible case is samples come from mixed distributions, e.g., mix of ImageNet-A&ImageNet-V2&ImageNet-R. I think the method will not work in such a circumstance.
    - 2.1 The proposed method will not work in an open scenario, for example, the number of classes is not fixed.

- 3. How about the inference time of the proposed method? The time cost of one step in Algorithm 1 from $for$ to $end for$.

**Questions:**

Please check Weaknesses.

---

> ### Author Response · Authors · 2023-11-18
>
> We value your constructive feedback and would like to address your comments with the following responses:
> 1. In light of the concern regarding ablation experiments on various components of BaFTA, we've introduced additional ablation results in the supplementary materials. Specifically, we've included BaFTA-single, utilizing a single-template CLIP as the base model, and BaFTA-Avg, which employs a simple average function to merge predictions. Notably, the results from BaFTA-Avg exhibit a 1.68% decrease compared to the regular BaFTA, underscoring the effectiveness of Renyi Entropy Aggregation. For a comprehensive exploration of these results, please refer to Section C.1, Section C.2, and Table 3 in the supplementary materials.
> 2. The consistent decent zero-shot classification accuracy achieved by CLIP across ImageNet and ImageNet-A/V2/W/R/Sketch using the same text embeddings generated from the CLIP text encoder suggests that the distributions of these datasets are not distinctly different from each other. Notably, ImageNet-R itself comprises a dataset with mixed distributions, including art, sketch, cartoon, plastic toys, etc. BaFTA demonstrates a significant improvement even in this mixed-distribution scenario. This indicates that BaFTA can effectively handle a certain level of distribution mixture, given that CLIP embeddings being sufficiently accurate, thereby ensuring that the distributions are not too dissimilar. It is possible to improve BaFTA with dynamic clustering mechanisms to handle an indefinite number of clusters or mixtures of severely different distributions. While this modification is beyond the scope of the current paper, we appreciate the suggestion and will consider it in future work.
> 3. In response to the request for an inference time analysis, we have included a comparison of the inference time for BaFTA and TPT in our supplementary materials. The results indicate BaFTA is roughly 5 times faster than TPT. For detailed results and analysis, please refer to Section A and Table 1.

---

> > ### Comment · Reviewer_sCvw · 2023-11-23
> > **Response**
> >
> > Thanks for the author's response.
> >
> > It seems that the proposed method is a mixture of online clustering+visual text alignment+aggregation with renyi entropy. After checking the newly provided Table 3 in supplementary_material. It seems aggregation with renyi entropy makes a significant contribution in the performance. However, I am not sure why the authors use such an aggregation method and how/why it improves the results.
> >
> > Besides, the mixture of different methods also makes me confused about the main contribution of the method (Reviewer JA78 points out that clustering appeared in previous works). Thus I think a more dedicated declaration about the contributions is needed.
> >
> > Overall, I maintain my original score, marginally below.

---

> > > ### Author Response · Authors · 2023-11-23
> > > **Clarification on motivation and contribution**
> > >
> > > Thank you for your valuable comments. We appreciate the opportunity to clarify the motivation and contributions of our work, BaFTA.
> > >
> > > **Brief Response:**
> > >
> > > **Simple online clustering does not yield consistent favorable results due to unbalanced clusters. Therefore, we are motivated to use Renyi Entropy to filter the reliable results from the noisy online clustering predictions to enhance final results. For a dedicated declaration on our contributions, please refer to the full response and the contribution summary paragraph.**
> > >
> > > **Full Response:**
> > >
> > > Both ReCLIP and BaFTA are inspired by the observation that CLIP's visual embeddings often exhibit meaningful clusters. As a result, both methods leverage this inherent clustering information to enhance predictions through unsupervised learning. In ReCLIP, the use of label propagation algorithm for assigning pseudo labels is not only time-consuming but also demands a significant number of samples to perform effectively and generate high-quality pseudo labels for self-training. We argue that such conditions may not be practical in many real-world applications. As a result, we are motivated to develop a test-time adaptation algorithm that can operate online, providing a more efficient and adaptable solution.
> > >
> > > However, naive online clustering algorithm struggles to consistently yield favorable results, especially on datasets with a large number of categories such as ImageNet. In our analysis of the suboptimal online clustering outcomes, we identified a significant correlation with unbalanced clustering assignments. Due to data and model biases, certain categories might get assigned with few or no examples, leading to low-quality and biased centroids. However, such impact of unbalanced clustering is typically confined to a small subset of categories, and the prediction results over unaffected categories are often satisfactory.
> > >
> > > **This motivated us to seek a method to distinguish between high-quality and poor-quality predictions generated by online clustering.** After experimenting with various options (as demonstrated in Table 7 of the main paper), we found that Renyi Entropy serves as an effective estimator of prediction quality. It enables us to filter out the high-quality predictions from online clustering, improving the overall prediction performance. Moreover, Renyi Entropy not only filters out high-quality online clustering predictions but also effectively finds high-quality predictions from various augmented views. As a result, BaFTA utilizes Renyi Entropy as a unified method to aggregate predictions based on their reliability estimated through Renyi Entropy.
> > >
> > > As shown in Table 3 from the supplementary materials, BaFTA exhibits a 1.68% improvement over BaFTA-Avg, underscoring the effectiveness of Renyi Entropy Aggregation. This enhancement stems from two key aspects: 1) Renyi Entropy enables BaFTA to selectively concentrate on high-quality predictions from online clustering, allowing it to extract valuable results from the noisy online clustering predictions; 2) Renyi Entropy facilitates a more effective combination of predictions from various augmented views, as illustrated in Table 7 of the main paper.
> > >
> > >
> > > **Our contributions can be summarized as follows:**
> > >
> > > 1. We propose to use online clustering to exploit the inherent relationships among visual embeddings of test examples, enhancing test-time adaptation results. As evidenced in Table 3 of the supplementary materials (as well as Table 5 from the main paper), BaFTA exhibits a 1.38% improvement over BaFTA-RA, underscoring the efficacy of online clustering.
> > > 2. We propose to use Renyi Entropy to aggregate noisy predictions from online clustering as well as from augmented views. As shown in Table 3 from the supplementary materials, BaFTA has 1.68% improvement over BaFTA-Avg, which indicates the effectiveness of Renyi Entropy aggregation.
> > > 3. In light of these two innovations, we introduce BaFTA, a backpropagation-free test-time adaptation algorithm. BaFTA effectively enhances CLIP by 3.80% across 15 datasets during inference, without requiring any labeled examples or offline training. BaFTA achieves state-of-the-art performance without relying on backpropagation training. This not only results in faster operation (5 times faster than TPT) but also mitigates the risk of potential model collapse, a challenge faced by many test-time adaptation algorithms.

---

> > > > ### Comment · Reviewer_sCvw · 2023-11-23
> > > > **Response**
> > > >
> > > > Thanks for the author's review.
> > > >
> > > > Well, it seems the choice of Renyi Entropy is vital for the whole performance. Some questions are raised here:
> > > > - TPT only makes predictions with a single input view, BaFTA uses aggregation of different data views. How about TPT with aggregations of different data views after its TTA process? Or, what is the performance of BaFTA without aggregation (only a single data view)?
> > > >
> > > > - I am still confused about why Renyi Entropy, and many other entropies can be applied. The authors also list other choices in Table 7. Simple entropy as weight is close to Renyi Entropy, by the way, what is the performance of simply using entropy as weights in Table 3? Plus, the authors say they are motivated by [1]. [1] also mention other entropy choices. The key question is, why Renyi Entropy and how it solves the problems.
> > > >
> > > > - **After re-checking the method, I think the ensemble of different predictions of various data views is vital for the whole performance.** However, this may not be fair for other methods like TPT as they only use the prediction of a single view as the output. So TPT+aggregation-of-different-data-views maybe necessary for us to figure out the effectiveness of online clustering and visual-text alignment. This is something I did not note when I first saw the paper.
> > > >
> > > >
> > > > [1] Aleksandr Laptev and Boris Ginsburg. Fast entropy-based methods of word-level confidence estimation for end-to-end automatic speech recognition. In 2022 IEEE Spoken Language Technology
> > > > Workshop (SLT), pp. 152–159. IEEE, 2023.

---

> ### Author Response · Authors · 2023-11-23
>
> Hi! Thanks for the reply. Since it is approaching the end of the rebuttal period, we will focus on sharing our insights in this response. Hopefully these answers will help you better understand BaFTA!
>
> **Augmentation Views**: Although TPT only uses a single view for prediction, it requires the computation of augmented views for its test-time training before inference for every example.
> BaFTA does not introduce extra computation or memory cost on augmentations in comparison with TPT. Both TPT and BaFTA require augmentations as essential components to enhance performance, while TPT utilizes them during test-time training, BaFTA directly utilizes them during inference.
>
> **Effectiveness of Online Clustering**: The effectiveness of online clustering can be evidenced by results from Table 5 from the main paper. BaFTA-RA uses $p_i$ for prediction (Renyi Entropy aggregated CLIP predictions on augmented views), and BaFTA uses $p_i + \hat{p_i}$ for prediction (Renyi Entropy aggregated CLIP predictions and Online Clustering predictions, on augmented views). BaFTA exhibits a 1.38% improvement over BaFTA-RA, underscoring the effectiveness of online clustering predictions ($\hat{p_i}$).
>
> **Why Renyi Entropy**: Entropy functions are good estimators of prediction confidence. For example, an uncertain uniform distribution $p_u = [1/k, 1/k,..., 1/k]$ will have Renyi Entropy $Re(p_u)=-1$ and a confident one-host distribution $p_c = [1, 0, 0, 0, 0]$ will have Renyi Entropy $Re(p_c)=0$. As a general form of entropy functions, we can control the sharpness of Renyi Entropy curves for it to mimic the behaviour of other entropy functions by tuning its entropy order $\alpha$. Small $\alpha$ corresponds to more sharpened entropy curves where only extremely confident predictions get high outputs. According to our ablation results in Section B of the supplementary materials, we find $\alpha=0.5$ reaches a good balance for predictions generated by CLIP embeddings, and therefore we use Renyi Entropy with $\alpha=0.5$ for prediction aggregation.

---

> > ### Comment · Reviewer_sCvw · 2023-11-23
> > **Response**
> >
> > Thanks for the author's response.
> >
> > Now, the key question becomes **"whether the ensemble of different data views plays a vital role (or the most important role)"**.
> >
> > From Table 5, we can see that **BaFTA-RA**(ensemble of different data views) achieves huge improvement compared to the baseline (**57.87 vs 49.89**) on ImageNet-A. It is close to the final result **58.19**.
> >
> > I also run the ensemble of different data views for TPT (without CoOp weights) on ImageNet-A. Following the TTA process of TPT, I choose the confident samples and simply average them and get results in the below table.
> > | Method | ImageNet-A |
> > |---|---|
> > | zero-shot CLIP-ViT-B/16 | 47.87 |
> > | TPT | 54.77 |
> > | BaFTA | 58.19 |
> > | TPT + ensemble-of-confident-data-views | **59.76** |
> > The **TPT+Ensemble-of-data-views** will be better if we also use some other entropies as weight scores.
> >
> > The above experiments and Table 5 in the paper both demonstrate "**the ensemble of different data views plays a vital role (or the most important role)"**.
> >
> > This does not flaw the other merits of the paper, for example, training-free. However, the arguments about online clustering/visual-text alignments/renyi entropy aggregation need further justification.
> >
> > Overall, I think a score of marginal below is suitable for the paper from my perspective.

---

### Official Review · Reviewer_mkqJ · 2023-10-31

**Soundness:** 3 good
**Presentation:** 3 good
**Contribution:** 2 fair
**Rating:** 6
**Confidence:** 2

**Summary:**

This paper addresses the challenge of enhancing zero-shot classification in vision-language models (VLMs) during inference. It introduces a novel approach called Backpropagation-Free Test-time Adaptation (BaFTA) that significantly improves zero-shot classification capabilities without requiring labeled examples or backpropagation training. BaFTA leverages online clustering algorithms to directly refine class embeddings within the unified visual-text space of VLMs. The authors also propose a method to combine clustering-based predictions with those from augmented views and evaluate their reliability using Renyi entropy. Comprehensive experiments validate BaFTA's effectiveness in enhancing zero-shot classification accuracy during inference.

**Strengths:**

+This paper excels in originality by introducing the Backpropagation-Free Test-time Adaptation (BaFTA) algorithm, a novel approach to enhancing zero-shot classification in vision-language models without labeled data or backpropagation.
+ The quality is evident through comprehensive experiments and the sound methodology of the online clustering technique.
+ Clarity is maintained in the well-structured introduction and clear explanations of the methodology.
+ The significance of the paper lies in its practical relevance, potential wider applicability, and the introduction of a technique for dynamically aggregating predictions using Renyi entropy, enhancing the robustness and accuracy of predictions in machine learning tasks.

**Weaknesses:**

- Efficiency plays a pivotal role in test-time adaptation, as the ability to swiftly adapt to novel environments is of paramount importance. The paper should explicitly report and compare inference time metrics for BaFTA with existing methods like TPT. This can be done by measuring the time required for BaFTA to adapt to new data during inference and comparing it with the time taken by TPT or other relevant methods. Providing quantitative results and possibly a comparison table would be valuable for the readers.
- The paper assumes that the visual embeddings from CLIP are often discriminative enough for effective classification. A more detailed discussion or evidence regarding this assumption would strengthen the paper's argument.
- BaFTA's success seems to depend on parameters such as the projection space and the Renyi entropy threshold. The paper could provide more insights on how these parameters are chosen, tuned, and their sensitivity to results. A sensitivity analysis or parameter ablation study could be included.

**Questions:**

Please refer to weaknesses.

---

> ### Author Response · Authors · 2023-11-18
>
> Thank you for your valuable comments! Here are our responses to your questions:
> 1. In response to the request for an efficiency analysis, we have included a detailed comparison of the inference time for BaFTA and TPT in our supplementary materials. Notably, BaFTA demonstrates an approximately 5 times speed improvement over TPT. For detailed results and analysis, please refer to Section A and Table 1.
> 2. Regarding the assumption about CLIP visual embeddings, results in Table 2 of the main paper indicate that CLIP's linear evaluation results on all datasets are significantly higher than the corresponding zero-shot results. Linear evaluation is a protocol that assesses the quality of representations.  By freezing the CLIP visual embeddings and training a fully-supervised linear layer classifier, linear evaluation obtains the upper bound accuracy achievable by the frozen visual embeddings. Table 2 results suggest that the CLIP performance is likely constrained by the quality of text-generated classification weights. While CLIP visual embeddings are not perfect, we argue that, in the unsupervised test-time adaptation setting where only minor modifications can be made, focusing on text-side updates is more efficient due to the substantial improvement space available on the text side.
> 3. To offer more insights into the sensitivity of the Renyi Entropy parameter, we have added updated plots and results in the supplementary materials. The analysis suggests that BaFTA's performance is not highly sensitive to the choice of $\alpha$. Please refer to Section B, Figure 1, and Table 2 for a more detailed analysis. In terms of the embedding projection, it does not involve additional hyper-parameters except for the number of basis used to construct the projection matrix. For the majority of datasets, we set this number equal to the number of classes, considering it as the rank and the maximum number of basis for the text embedding subspace. However, for datasets with more than 150 categories, such as SUN397 and ImageNet, we observed that utilizing the top 150 basis is sufficient to capture most important dimensions from the embeddings. Therefore, we use the top 150 basis for SUN397 and ImageNet, as mentioned in the implementation detail section.

---

> ### Comment · Reviewer_mkqJ · 2023-11-20
>
> Thank you for addressing my main concerns. I will vote for weak accept.

---

> > ### Author Response · Authors · 2023-11-23
> >
> > Thank you!

---

### Official Review · Reviewer_mwjV · 2023-11-10

**Soundness:** 3 good
**Presentation:** 3 good
**Contribution:** 3 good
**Rating:** 5
**Confidence:** 3

**Summary:**

The paper discusses the challenges faced by test-time prompt tuning methods in enhancing the performance of large-scale pretrained vision-language models. The authors a novel backpropagation-free method for test-time adaptation in vision-language models, focusing on estimating class centroids through online clustering in a projected embedding space that aligns text and visual embeddings. The approach dynamically aggregates predictions from both estimated and original class embeddings, as well as distinct augmented views, while assessing prediction reliability using Renyi entropy. The authors claim that the proposed method reaches the state-of-the-art test-time adaptation methods significantly.

**Strengths:**

- The authors use of a backpropagation-free approach instead of employing prompt-tuning is novel than before approach.

- The main contributions are clearly stated.

**Weaknesses:**

- The experimental setting parameters are not the same with TPT and no fair experiment is conducted. The paper of Implementation Details shows: In line with the TPT implementation, we utilize a simple combination of RandomResizedCrop and RandomFlip to prepare 63 augmented views, constituting a mini-batch of 64 images for each test image. Actually, TPT only uses random resized crops to augmented views. Besides, “and adaptation In line with the TPT” seems less a “.”

- Although the authors' approach can use multiple prompts compared to TPT only use single prompt, the paper lacks ablation experiments on prompt

- Authors can present results in more forms of Effectiveness of Projected Embedding Space, such as t-SNE.

**Questions:**

Although the authors use of a backpropagation-free approach instead of employing prompt-tuning is novel, some experiments are still needed to prove the effectiveness of the method.

---

> ### Author Response · Authors · 2023-11-18
>
> Thank you for your valuable comments! Here are our responses to your questions:
> 1. Regarding the implementation details, we carefully followed the official implementation from the [GitHub Repository](https://github.com/azshue/TPT/tree/main) provided in the TPT paper. In the actual implementation of TPT, both RandomResizedCrop and RandomFlip augmentations were used to produce the reported scores. Therefore, the comparison of BaFTA and TPT is fair.
> 2. In response to the concern about ablation experiments on prompts, we evaluated BaFTA with single-template CLIP as the base model. The BaFTA-single configuration resulted in a noteworthy 6.22% improvement in averaged accuracy across 15 datasets. You can find detailed results and analysis in Section C.1 and Table 3 of the supplementary materials.
> 3. To provide a more comprehensive understanding of the effectiveness of the projected embedding space, we have included t-SNE plots of CLIP visual embeddings in the supplementary materials. The t-SNE plot visually demonstrates that the projection space can effectively transform the sparse clusters into denser formation. Please refer to Section E and Figure 2 for detailed results and analysis.

---

### Author Response · Authors · 2023-11-18

Thanks to all the reviewers for their constructive comments! In this revised version, we've carefully addressed a few typos in the main paper and presented additional results in the supplementary materials in response to the insightful questions raised during the rebuttal.

Notable additions include new ablation studies on templates (prompts), a detailed exploration of Renyi Entropy aggregation, a sensitivity analysis on Renyi Entropy order $\alpha$, illustrative tSNE plots showcasing the effectiveness of the projection embedding space, and an insightful inference time analysis, which shows BaFTA is approximately 5 times faster than TPT. For a comprehensive understanding of these updates, please refer to the supplementary materials.

We extend our gratitude for the positive feedback on our method's simplicity, practical relevance, wider applicability, and novel approach. The remarks on the excellence in originality and the clarity of our paper writing are deeply appreciated.

Thank you once again for your valuable insights.

---

> ### Author Response · Authors · 2023-11-23
>
> Fixed some more typos in the main paper.

---

### Meta-Review · Area_Chair_zAfX · 2023-12-31

**Metareview:**

The paper addresses the problem of test time adaptation of pre-trained vision models in a zero-shot classification setting. The focus is on a back-prop free approach that instead relies on clustering.

Initial reviews commended the interesting and novel approach and highlighted the strong empirical performance. At the same time, a number of questions were raised, primarily to further understand the approach, how it works, and insights that can be gained from the present work.

Substantial follow up discussion between reviewers and authors resulted in clarification of some of the initial questions and concerns. The authors provided additional empirical results and analysis.

At the present state, reviewers feel more confident that they understand the components of the method proposed here. Substantial new insights have been produced during the rebuttal and the precise role played by each of the components of the proposed approach have been better understood. Reviewers and AC feel that these insights must be fully integrated and validated in an updated manuscript - to fully address the question of how each of the proposed components contributes to the observed empirical performance. As a result the recommendation is to reject the manuscript in its present form.

**Justification For Why Not Higher Score:**

Empirical results and computational benefits are strong. At the same time, the proposed method consists of multiple components and the contribution of each needs to be understood through thorough analysis.

**Justification For Why Not Lower Score:**

N/A

---

### Decision · Program_Chairs · 2024-01-16

Reject